

# A global database of historic glacier lake outburst floods

Natalie Lützow[1], Georg Veh[1], and Oliver Korup[1,2]

[1]Institute of Environmental Science and Geography, University of Potsdam, Potsdam-Golm, 14476, Germany
[2]Institute of Geosciences, University of Potsdam, Potsdam-Golm, 14476, Germany

*Correspondence to*: Natalie Lützow (natalie.luetzow@uni-potsdam.de)

**Abstract.** Ongoing atmospheric warming has accelerated glacier mass loss in many mountain regions worldwide. Glacier lakes trap parts of the glacial meltwater and increased by about 50% in number and area since the 1990s. Some of these glacier lakes may empty catastrophically and pose hazards to mountain communities, infrastructure, and habitats. Such glacier lake outburst floods (GLOFs) have caused millions of dollars of damages and fatalities, and are one of many concerns about future

changes in the magnitude, frequency, and impacts of a shrinking mountain cryosphere. Consistently compiled inventories are thus vital to assess regional and local trends in GLOF occurrence, hazard, and risk. To this end, we studied 769 literature and internet sources, and developed a standardised database with 57 parameters that describe and quantify the location, dam type, size, timing, and impacts of GLOFs in nine glaciated mountain regions. Our GLOF inventory also includes details about the lake area before and after the outburst for 391 cases that we manually mapped from optical satellite images since 1984. In

total, we compiled 3,151 reported GLOFs that occurred in 27 countries between 850 and 2022 C.E. Most GLOFs have been reported in NW North America (26%) and Iceland (19%). However, the reporting density in our inventory varies. During the 20[th] century alone, the number of yearly documented GLOFs increased 6-fold. Less than one-quarter of all reported cases feature hydrodynamic characteristics such as flood peak discharge or volume, or estimates of loss and damage. Our inventory more than doubles the number of reported GLOFs in a previous global inventory, though gaps in attributes remain. Our data

collection process emphasizes the support of local experts in contributing previously undocumented cases, and we recommend applying systematic protocols when reporting new cases. The global database on historic GLOFs is archived at https://doi.org/10.5281/zenodo.7330345 and regularly updated at http://glofs.geoecology.uni-potsdam.de/.

## 1 Introduction

Population growth and socio-economic development have increased the exposure of people and infrastructure to natural

hazards in high mountain areas (Hock et al., 2019). These regions host glaciers and permafrost and are highly sensitive to rapidly rising air temperatures in past decades (Koven et al., 2013; Hugonnet et al., 2021). While the resulting losses of ice are one symptom of a dwindling cryosphere, further consequences include changes to stream flow seasonality, decreased freshwater availability and hydropower potential, and potential socio-economic tension along rivers originating from glaciated high mountains (Kääb et al., 2005; Hock et al., 2019; Pritchard, 2019). Fed by ongoing glacier retreat, large masses of





meltwater can be temporarily trapped in deglaciated depressions or at the margins of glaciers. Between 1990 and 2018, global glacier lake volume increased by around 48% to 156.5 km³, and lake abundance increased by 53% (Shugar et al., 2020). Stable and long-lived lakes can become viable resources for drinking water and irrigation (Farinotti et al., 2019). However, some lakes have unstable dams and may pose hazards to downstream communities if failing suddenly (Lütschg, 1915; Mathews and Clague, 1993; Clague and Evans, 2000). Hundreds of such Glacier Lake Outburst Floods (GLOFs) have been observed in

glaciated mountain regions in past centuries (**Fig. 1**). GLOFs are sudden pulses of water and debris from a meltwater body, and can disturb river reaches tens to hundreds of kilometres downstream from their sources with high damage potential (Kääb et al., 2005; Carrivick and Tweed, 2016; Hock et al., 2019). The high hydraulic head of the flood source promotes high flow velocities and peak discharges, in many cases similar or higher than meteorological floods (Cenderelli and Wohl, 2001; Emmer, 2017; Jacquet et al., 2017; Cook et al., 2018). Ecologic and geomorphic disturbances from GLOFs range from eroded

vegetation along river banks to alluvial fans and deltas with thick deposits of woody debris and sediments, destroyed or debris-covered forests, and impacts on aquatic wildlife, for example by killing fish populations or hindering salmon spawning (Cenderelli and Wohl, 2001; Otto, 2019; Pitman et al., 2021; Geertsema et al., 2022). In populated areas, GLOFs can destroy houses, roads, and bridges as well as farmland and livestock, causing millions of dollars of losses for the local economy and property (Carrivick and Tweed, 2016). Owing to their short warning times and limited monitoring capabilities, GLOFs have

caused many fatalities, such as during the failure of Cirenmaco (Zhangzangbo) lake in 1981. This particular outburst flood alone killed more than 200 people (Xu, 1988; Shrestha et al., 2010; Wang et al., 2018).
    Despite growing evidence of more and larger historic glacier lakes, estimates of whether GLOF hazard and risk have increased commensurately remain controversial (Richardson and Reynolds, 2000; Harrison et al., 2018; Nie et al., 2018; Veh et al., 2020; Bazai et al., 2021; Stuart-Smith et al., 2021; Compagno et al., 2022). We find that GLOF reporting has been inconsistent across

different countries and study regions without any standardised procedures (Carrivick and Tweed, 2016). Information on GLOF timing, magnitude, and impacts varies between case studies, reflecting the different focus on either hydrodynamic flood characteristics (Osti and Egashira, 2009), geomorphic changes (Tomczyk et al., 2020), or societal impacts (Carey, 2005). The few available regional inventories also cover different time intervals. Hence, appraisals of regional GLOF trends remain challenging. Scientists can draw on only two published global catalogues by Carrivick and Tweed (2016) and Harrison et al.

(2018), and the latter focuses on GLOFs from moraine-dammed lakes only. Both studies discussed issues of incomplete, non-systematic GLOF monitoring and reporting, and called for standardised procedures and data sharing. Here we meet this demand for a systematic documentation and pave the way to objectively monitor GLOFs occurrence, hazard, and risk (Kääb et al., 2005; Emmer et al., 2016; Emmer, 2017). Our goal is to present a global database that covers consistently and systematically the largest number of reported GLOFs.



## 2 Methods

### 2.1 Data sources

We reviewed a total of 769 resources to collect information on historic (i.e. documented) GLOFs in nine major glaciated regions: the Andes (i.e., Argentina, Bolivia, Chile, and Peru), NW North America (i.e., western Canada and the western USA), Greenland, Iceland, Scandinavia (i.e., Norway and Sweden), the European Alps (i.e., Switzerland, Italy, and Austria), the Caucasus (i.e., Georgia and Russia), High Mountain Asia (i.e. Bhutan, China, India, Kazakhstan, Kyrgyzstan, Tajikistan, Afghanistan, Nepal, and Pakistan), and New Zealand. The Caucasus and New Zealand had few reported cases, and are thus part as region 'Other' in the following. Our study regions are similar in extent to those of the major regions in the Randolph Glacier Inventory (RGI; RGI Consortium, 2017, http://www.glims.org/RGI/, last accessed 29 October 2022) (**Fig. 1**).

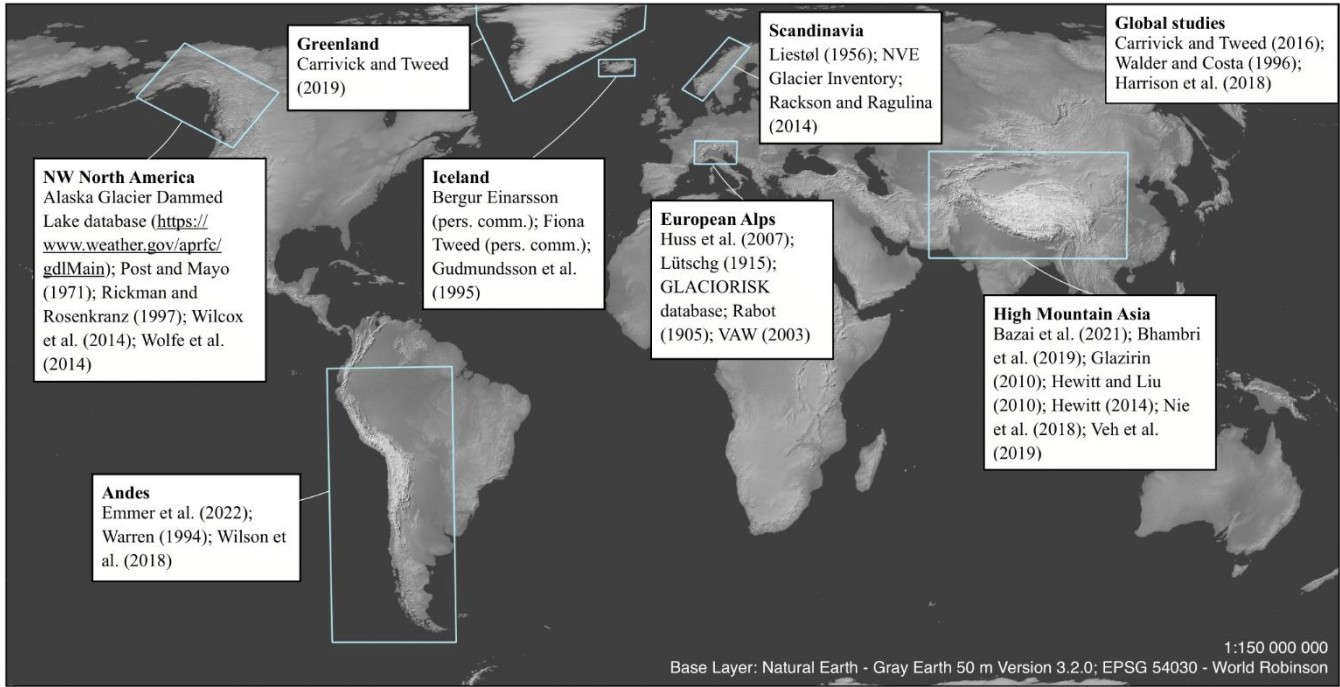

**Figure 1: Global map of the major study regions.** Boxes show sources that we regard as key sources in a given region.

We divided previous work in these regions into primary sources (55%), where we had direct access to the original report, and secondary sources (45%), such as summaries or reviews of historical cases without direct access to the original reference (**Fig. 2a**). Most sources are peer-reviewed publications (76%). The remainder includes non peer-reviewed literature (18%) such as official reports, local databases, and conference abstracts, as well as news and social media content (6%), including videos, blog posts, web articles, and newspaper articles (**Fig. 2b**). We also asked 14 researchers to review and contribute cases using their local knowledge (see Acknowledgements). Most of our sources are from High Mountain Asia ($n = 229$) and the European Alps ($n = 213$), while Greenland, Scandinavia, and the study region 'Other' had the lowest research output (**Fig. 2a**). The

European Alps are covered by a comprehensive database of glacier-related hazards arising from the GLACIORISK project

(http://www.nimbus.it/glaciorisk/gridabasemainmenu.asp, last accessed 27 October 2022), although we had limited direct

access to the literature for some of the GLOFs.

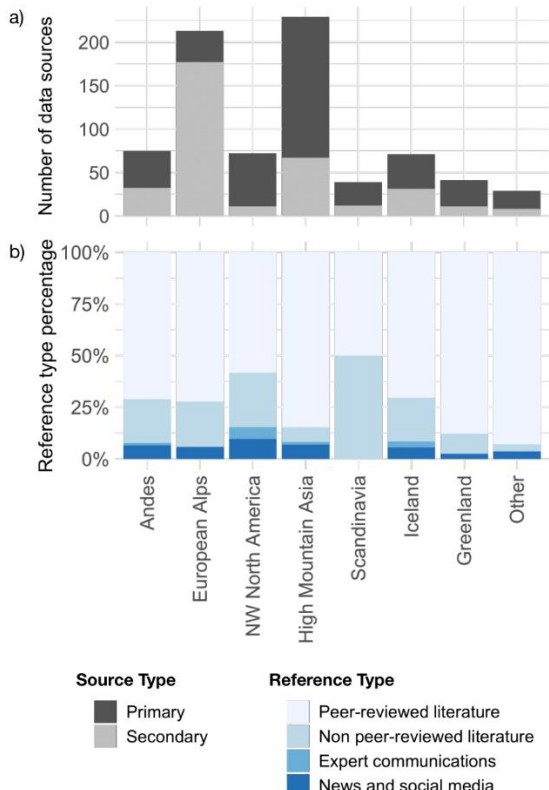

**Figure 2: Data sources of GLOFs by study region. a)** Number of primary and secondary data sources, and **b)** the proportion of the reference types in each region.

**2.2 Structure of the database**

We compiled 57 attributes to capture quantitative and qualitative information on location; timing; hydrodynamic

characteristics; socio-economic and geomorphic impacts; and references associated with all reported GLOF in our database

(**Fig. 3**). We summarize these attributes below:



**Figure 3: Structure of the GLOF database.** White boxes are input data. Black backgrounds show major categories of GLOF attributes, including finer branches that specify these parameters. Parameters with black fonts are gathered from the literature, parameters with blue fonts are derived from additional data. Data types and units of the parameters are given in square brackets; [i]: integer, [char]: character, [num]: numeric; [YYYY]: year; [MM]: month, [DD]: day.

**ID.** We separated the GLOFs by study regions and gave each case a unique identifier.






**Location.** We included eleven metrics describing each GLOF such as the geographic coordinates and the local name of the source lake, mountain range, country, glacier name, and the river impacted by the lake drainage, if available. We used the Randolph Glacier inventory Version 6.0 (RGI Consortium, 2017, http://www.glims.org/RGI/, last accessed 29 October 2022)

to add the ID and the area of the parent glacier as additional information. We also report or assess the type of dam impounding the lake, and the mechanism that led to drainage. We differentiated between lakes dammed by glacier ice, moraines, bedrock, or a combination thereof, as well as englacial water pockets, and subglacial lakes associated with geothermal or volcanic activity (Carrivick and Tweed, 2013; Emmer et al., 2022a). Bedrock-dammed lakes form in topographic depressions and overdeepenings, for example, by glacial erosion (U.S. Department of Agriculture, 2016). Moraine-dammed lakes trap water

behind lateral or terminal moraines (Otto, 2019). Ice-dammed lakes can be impounded by trunk glaciers in ice-free tributary valleys or by advancing glaciers that temporarily block surface runoff (Iturrizaga, 2011). Further types of ice-dammed lakes include supraglacial lakes (on top of glaciers), and water pockets within or beneath the glacier (Haeberli, 1983; Benn and Evans, 2010). We differentiated supraglacial lakes and water pockets from subaerial glacier-dammed lakes to emphasize differences in dam shapes and breach dynamics. Where unavailable in the literature, we used optical satellite imagery from

Google Earth Engine and digital topographic maps (OpenTopoMap; ESRI World Topo) to infer the type of dam.

**Date.** The reported date of each GLOF is essential to learn more about changes in GLOF occurrence and timing. Some sources offer a range of plausible dates, especially where the floods were determined by comparing two satellite images showing changes in characteristic geomorphic diagnostics of GLOFs such as a loss of lake area or sediment deposition downstream (Nie et al., 2018; Veh et al., 2019; Emmer et al., 2022a).

**Hydrodynamic GLOF characteristics.** The outburst mechanism controls the average and maximum flow rate, velocity, and volume, and thus the shape of the hydrograph. Different dam types allow for different shapes of the discharge path during outburst (Röthlisberger, 1972; Fowler, 1999; Westoby et al., 2014; Walder et al., 2015). Ice-dammed lakes may drain through one or more subglacial tunnels, flotation, incision, or failure of the dam (Walder and Costa, 1996; Otto, 2019). Some of these mechanisms allow ice-dammed lakes to fill and drain repeatedly, for example if subglacial tunnels open during and seal after

an outburst flood (Nye, 1976; Clarke, 1982). The stability of a moraine dam depends, among others, on the dam geometry, internal structure, and material properties (Weidinger et al., 2002; Korup and Tweed, 2007). Failure occurs when the strength of the material is exceeded, for example due to shear stresses because of seepage or an increased load on the dam due to rising lake levels (Korup and Tweed, 2007). Melting ice cores in moraine dams may decrease stability, leading to breach in some cases (Richardson and Reynolds, 2000). Overtopping is a commonly observed outburst mechanism for moraine- and bedrock-

dammed lakes and often a consequence of displacement waves from mass movements (ice, snow or rock avalanches) into the lake (Haeberli et al., 2017). Another mechanism leading to overtopping is excessive runoff due to intense rainfall, snowmelt or glacier recession (Costa and Schuster, 1988). Overtopping can destroy a moraine dam by eroding the dam through drainage channels (Costa and Schuster, 1988).

Lake volume ($V_L$), flood volume ($V_0$), and peak discharge ($Q_p$) are important baseline parameters to quantify GLOF magnitude.

Where studies provided ranges of these values, we reported minima, maxima, and the arithmetic mean. To acknowledge the



reliability of reported values of $V_0$ and $Q_p$, we distinguished between gauged, estimated, and unknown flood quantities. $V_0$ is considered as *gauged* if the flood volume was obtained from a hydrograph, and *estimated* if the lake had one or several bathymetric surveys before or after the flood.

We mapped GLOFs between 1984 and 2021 using multispectral optical images from eight satellite missions (Landsat 4, 5, 7, and 8, Geo-Eye 1, RapidEye, Sentinel-2, and Planetscope), due to limited operation times (**Table 1**). Ground sampling distance and revisiting times over a given location vary throughout the dataset. Overall, we obtained Landsat imagery from the Google Earth Engine Data Catalog (https://developers.google.com/earth-engine/datasets/catalog/landsat), and Planetscope, RapidEye, and Sentinel-2 imagery from the Planet Explorer operated by Planet Labs PBC (https://www.planet.com/products/explorer/). Most of the images ($n$ = 359) were from the Landsat-5 mission, which started in 1984 and ended in 2012. In each case, we selected images as recent as possible with respect to the outburst date, filtering to images with cloud cover of <20%, and minimum snow and shadow cover. For cases with known outburst date, the median time difference between image date and the outburst date is 14 days before the GLOF, and 20 days after the GLOF. We mapped lake area before ($A_b$) and after the GLOF ($A_a$) from 839 images for 32% of all GLOFs after 1984 (391 outbursts) as polygon features in QGIS V3.18. This allows users to calculate changes in lake area owing to the GLOF. Lake area is a rough proxy of the GLOF size because estimated values of $V_L$, $V_0$, and $Q_p$ often originate from different empirical scaling relationships (Clague and Mathews, 1973; Evans, 1986; Walder and Costa, 1996).

**Table 1: Resolution and source satellite missions of images used to map glacier lake areas before and after outbursts.**

| Satellite mission | Operation time | Pixel resolution [m] | Number of images |
|---|---|---|---|
| **Landsat 4** | 1982-1993 | 30 | 1 |
| **Landsat 5** | 1984-2013 | 30 | 359 |
| **Landsat 7** | since 1999 | 30 | 180 |
| **Geo-Eye 1** | since 2008 | 0,41 | 4 |
| **RapidEye** | 2008-2020 | 5 | 52 |
| **Landsat 8** | since 2013 | 30 | 109 |
| **Sentinel-2** | since 2015 | 10 | 9 |
| **Planetscope** | since 2016 | 3 | 125 |

The pixel resolution refers to the resolution of the spectral bands that we used to visually map lakes.

We also included in our database the lake outlines from 11 GLOFs mapped by Bazai et al. (2021) in the Karakoram, and nine GLOFs mapped by Eide (2021) in Scandinavia. We have not yet mapped lake areas in Greenland, but plan to include them in a future update of our database. We mapped the lake polygons in the local UTM projection using image enhancement such as clipping band histograms or changing image contrast and colour saturation. False-colour composites further increased the visual difference between vegetation, soil, water, and ice. We found the near infrared (NIR) bands useful to map lakes using colour composites such as [NIR1, red, green] or [NIR1, NIR2, red]. From the mapped lake polygons we calculated lake area

and perimeter with the statistical programming software R (https://cran.r-project.org/) using the st_area and st_perimeter
function included in the sf package in the local UTM projection (V1.0-7; https://github.com/r-spatial/sf/). For all mapped lake
areas, we included the name of the satellite scene, the image date, and the lake perimeter. We also added an estimate of
reliability for the mapped lake areas. We assigned "low confidence" to lake perimeters that were partially shrouded by ice,
shadows, or clouds, and "high confidence" to clear views onto the lake surface.

**Impacts.** We compiled and split reported flood impacts into ten categories to rank socioeconomic impacts. Our damage
categories expand on those of the global GLOF assessment by Carrivick and Tweed (2016). We distinguished between
economic and infrastructural losses, environmental impacts, and social consequences with regard to fatalities or a need for
resettlement of the affected population. Economic losses include, for example, damage to farmland or destruction of touristic
facilities. Furthermore, damages to infrastructure were separated into buildings, bridges, roads, railroads, and utilities such as
water and electricity supply and communication services. If available, we also documented the number of damaged features
of each category. Where information on GLOF impacts was vague, we distinguished between features that were damaged or
only affected without structural damage, for example by covering a road with debris.

**Comments.** We documented any additional information from our sources, such as extreme weather events involved in
triggering lake outbursts, the mean discharge during the GLOF, information on river stages, or anecdotal, non-numerical
descriptions of the course of the flood and its characteristics, such as flood duration, flow type or sediment load, or subjective
descriptions of the flood magnitude (e.g. minor, large, or extensive flood).

**References.** Finally, we listed all sources from which we extracted information on GLOFs. We highlighted earlier published
information that was cited in more recent publications by linking them with "CITED IN", independent of the accessibility to
the cited source. If we had access to a cited reference, we always searched for the original source to validate the provided
information. If a publication provided multiple sources for an event (e.g. in data tables), the cited references were connected
with an "&" operator. Single sources were separated by semicolons. Lastly, we noted the year of the first reference found that
cited the event.

## 3 Results

### 3.1 Spatial distribution of GLOFs

Our database has a total of 3,151 GLOFs that originated in 27 countries between 850 and 2022 C.E. (**Fig. 4**). With 833 reported
cases, NW North America is the region with the highest number of outbursts, followed by Iceland (*n* = 590) and High Mountain
Asia (*n* = 569*)*. Few GLOFs were reported in Scandinavia (*n* = 192) and Greenland (*n* = 153) (**Fig. 5b**). The Andes had the
highest number of outburst sources (*n* = 218), but only few more GLOFs (*n* = 337) (**Fig. 5a, b**). The database contains about
three times more GLOFs than GLOF source locations in High Mountain Asia, four times more in the European Alps, five
times more in Scandinavia, seven times more in Iceland, and nine times more in NW North America (**Fig. 5c**). Overall, 64%
of all individual source locations in the database had exactly one outburst, with up to 91% of single drainage events located in



the Andes. Most lakes with multiple outbursts were in NW North America, where 30 lakes produced at least five outbursts each.

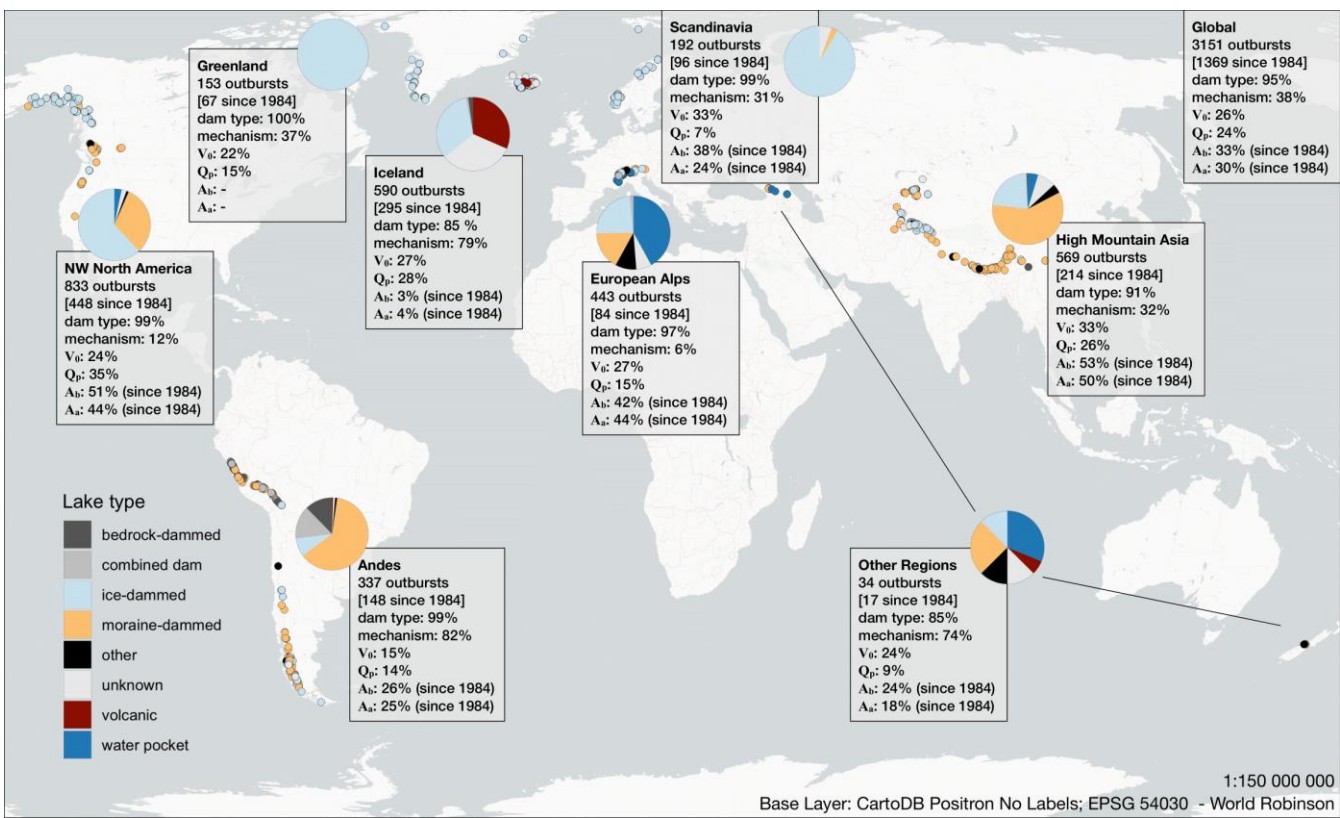

**Figure 4: Overview map of the source locations of GLOFs and selected regional database contents.** Colours show the lake type of the GLOF locations (bubbles) and proportions of GLOF locations in each region (pie charts). Boxes are regional statistics on the total number of outbursts [and the number of outbursts of GLOFs since the availability of satellite images]; the percentage of cases with reported dam type and drainage mechanisms; and parameters on GLOF magnitude such as peak discharge ($Q_p$), flood volume ($V_0$), and the area before ($A_b$) and after ($A_a$) the GLOF.

Dam type is one of the most consistently reported attributes and available for 95% of all cases. Most reported lake outbursts involved lakes dammed by ice including ice marginal lakes dammed by the glacier body (65%), ice-dammed lakes affected by geothermal or volcanic activity (8%), and englacial water pockets (5%). Failures of moraine-dammed lakes account for 13% of all documented outbursts. GLOFs from bedrock-dammed lakes and lakes with combined dams or other sources, such as supraglacial lakes, are rare in our database (4%). We note that this global distribution differs regionally. In NW North America and Scandinavia, most GLOFs originated from ice-dammed lakes (**Fig. 4**). In contrast, High Mountain Asia and the Andes had most outbursts from moraine-dammed lakes. Scandinavia had only one reported outburst from a moraine-dammed lake (Breien et al., 2008). In NW North America, every third outburst on average came from moraine-dammed lakes, mostly reported in British Columbia (McKillop and Clague, 2007). By contrast, outbursts from ice-dammed lakes prevailed in Alaska (Post and Mayo, 1971). The Andes are the only region with many known outbursts of bedrock-dammed lakes ($n = 27$). Water pockets



have been rarely reported to burst out in all regions except the European Alps, where they represent 49% of the reported

outburst sources. Few GLOFs ($n = 14$) from water pockets were also reported in the Russian part of the Caucasus and in New

Zealand (region 'Other'). Iceland is the only region with a high proportion of reported GLOFs from subglacial lakes linked to

geothermal or volcanic activity (35% of all Icelandic GLOFs). Another 36% of GLOFs in Iceland originated from subaerial

ice-dammed lakes dammed in tributary valleys.

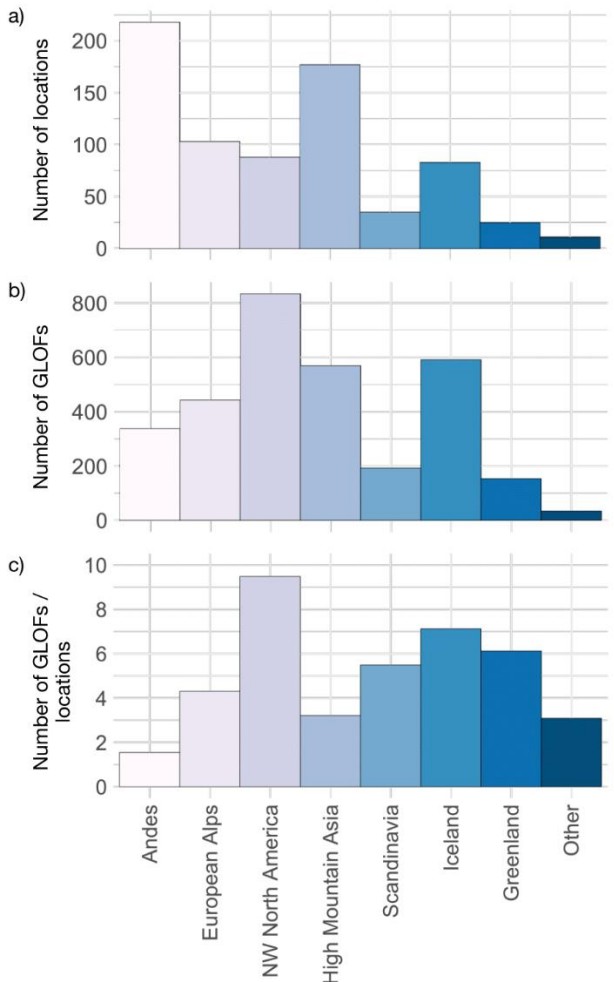

**Figure 5: Regional distribution of reported a) GLOF source locations, b) all GLOFs, and c) ratio of the number of**
**GLOFs and source locations.**

### 3.2 Temporal distribution of GLOFs

Our database spans more than 12 centuries. The earliest GLOF occurred in the year 850, where a volcanic eruption caused a

subglacial lake outburst at Mýrdalsjökull, Iceland (GLACIORISK, http://www.nimbus.it/glaciorisk/gridabasemainmenu.asp,

last accessed 27 October 2022). Before 1900, reporting activity was low with 371 cases, i.e. 12% of the total count (**Fig. 6**).

About half of all GLOFs before 1900 were reported in the European Alps where research interest had been high, judging from



many local archives, chronicles, and surveys about glacier changes and hazards (De Tillier, 1738; Baretti, 1880; Forel et al., 1882; Richter, 1892). In contrast, the Andes had only five reported GLOFs before 1900. Between 1900 and 2022, the decadal average of annually reported GLOFs increased 6-fold. Until the 1970s, the average increased with a rate of about five GLOFs per decade (**Fig. 6**). Between the 1970s and the 2000s, the increase in the reporting rate lowered to about one to two more

reported annual GLOFs per decade, followed by an increase of the decadal average by 14 GLOFs between the 2000s and the 2010s. About 39% of all outbursts took place after the start of the Landsat 5 mission in 1984.

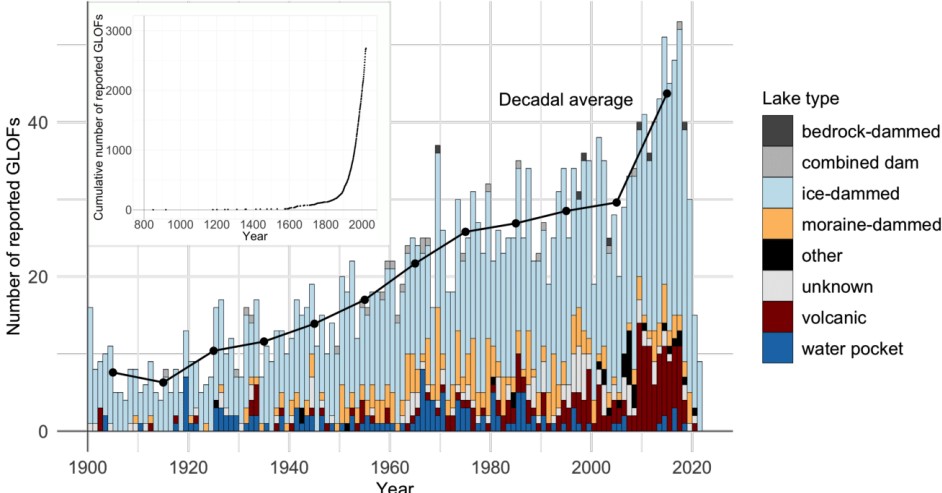

**Figure 6: Global annual number of reported GLOFs since 1900.** Colours distinguish the dam types. Inset shows the cumulative number of reported GLOFs since 850.

GLOFs from ice-dammed lakes have been reported yearly since 1900 and also most frequently compared to all other lake

types. However, moraine-dammed lake outbursts were also observed annually after 1950 with four exceptions (1996, 2007, 2021, 2022). Outbursts associated with volcanic activity and water pockets were reported less regularly, especially between 1900 and 1930. From 2010 to 2019, draining lakes associated with volcanic activity had a higher proportion of the total annual GLOF count compared to the rest of the observation period (**Fig. 6**). In contrast, GLOFs from bedrock-dammed lakes are only reported for a few years, all after the 1960s.

**3.3 Hydrodynamic GLOF characteristics**

Information on the outburst mechanism is available only for 38% of all cases, and even fewer GLOFs have estimated or gauged peak discharges $Q_p$ (24%) or flood volumes $V_0$ (26%). The data density of $Q_p$ or $V_0$ changes with time. Before 1900, only 9% of the reported GLOFs had one of these diagnostics described. The relative proportion of available data has increased since, especially between 1900 and 1950 (**Fig. 7**). After 1950, the proportion of available data levelled out to about one third until

the end of the observation period. Data density also differs between the study regions (**Fig. 4**). More information on $Q_p$ is available in NW North America, High Mountain Asia, and Iceland (~ 30%), as opposed to Scandinavia, the Andes, Greenland, and the European Alps (<15%). Data on $V_0$ are available for 20-35% of GLOFs in all regions except the Andes (15%). We



were able to map lake areas for about half of the reported outbursts in High Mountain Asia, NW North America, and the
European Alps since 1984. Due to high cloud cover, lake area could only be mapped for 4% of the outburst in Iceland since
1984. The outburst mechanism is reported for most GLOFs in Iceland and the Andes, but only few GLOFs in NW North
America and the European Alps.

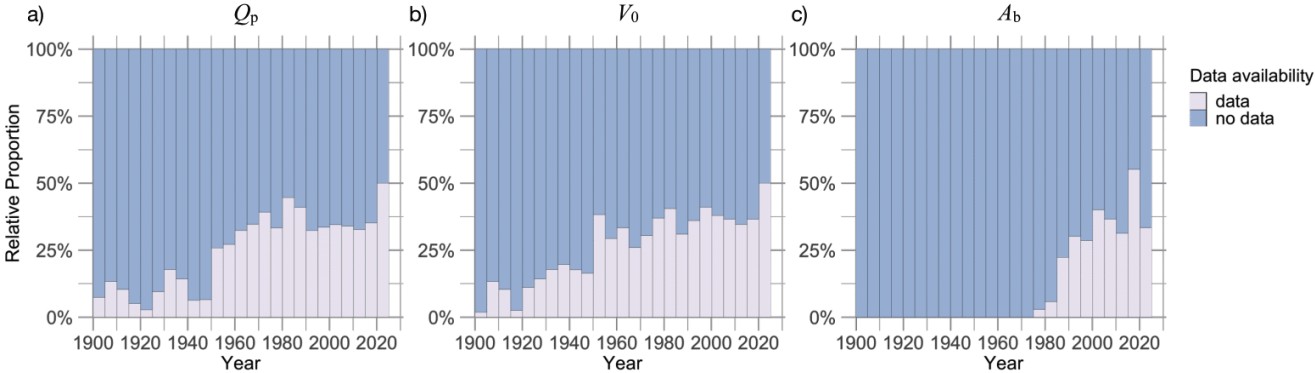

**Figure 7: Proportion of GLOFs with reported values of a) peak discharge $Q_p$, b) flood volume $V_0$, and c) the lake area
before the GLOF $A_b$ in five-year bins.** Consistent values for $A_b$ became available with the start of the Landsat era in the
1980s.

**3.4 GLOF impacts**

Flood damages are mentioned for 404 GLOFs. Almost half of the GLOFs with reported damages were associated with ice-
dammed lakes (49%), followed by moraine-dammed lakes (20%), and water pockets (17%) (**Fig. 8**). The most commonly
reported impacts were destroyed bridges ($n = 248$), economic losses ($n = 127$), and damaged or debris covered roads ($n = 104$). Most GLOFs that caused economic losses or damage to bridges, buildings and roads, originated from ice-dammed lakes
(**Fig. 8**). Our data contain 44 deadly GLOFs, 29 of them with a reported number of victims, six known to have killed more
than 100 people each. Many sources remained vague or offered estimates about the number of fatalities (e.g. Fushimi, 1985;
Fort, 2015), mostly due to missing information. At least 33 GLOFs caused damage to utilities, for example by cutting off or
shortening the local water supply, destroying pipes, or causing damage to hydropower plants. Most of the GLOFs that caused
damage to utilities originated from moraine-dammed lakes (**Fig. 8**).





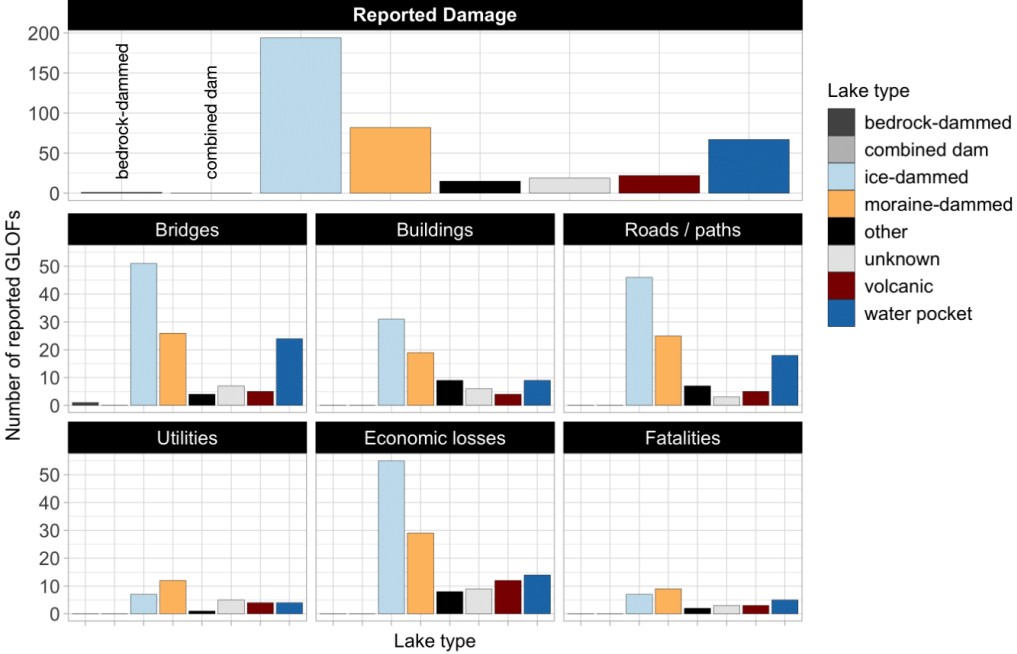

**Figure 8: Number of GLOFs with reported damages sorted by lake type.** Top row shows the total amount of reported damages for all lake types. Smaller panels show the number of reported GLOFs that caused damage to bridges, buildings, roads, utilities, economic losses, or fatalities.

76% ($n = 309$) of those GLOFs with reported damages occurred after 1900. However, between 1900 and present, only 12% of all reported GLOFs ($n = 2,484$) had damages. The decadal average of annually reported GLOFs that caused damage increased 3-fold since, from about two GLOFs in the first decade of the 20th century to about six in the 2010s (**Fig. 9a**). Only 21 (17%) of all years since 1900 remained without information on losses or damages, and since 1950, only five years had no noteworthy GLOF impacts. The ratio between annually occurring GLOFs with reported damages and the total number of reported events varied only slightly throughout the 21st century (**Fig. 9b**). However, in the first half of this century, the ratio was >0.2 in about 60% of years with reported GLOFs, which applies to only three years (5%) with reported GLOFs since 1950.

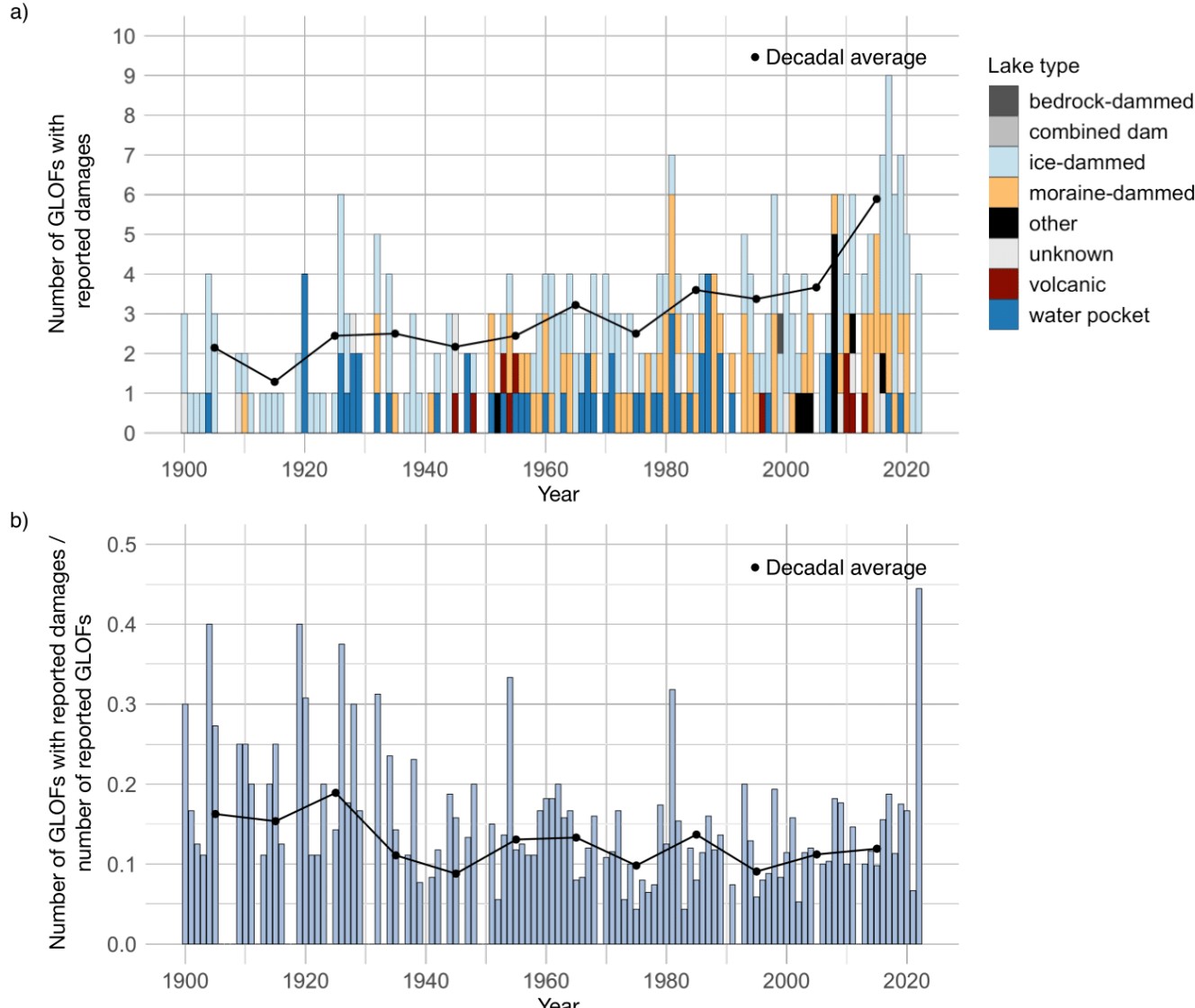

**Figure 9: Temporal distribution of GLOFs with reported damages (1900-2022). a)** Annual number of GLOFs with reported damages by lake type. **b)** Fraction of yearly reported GLOFs with details of damages. The black line is the decadal average of the annual count.

# 4 Discussion

## 4.1 Comparison with previous global databases

Our database is an important step towards approximating the historic and current frequency and size of GLOFs. We reviewed and compiled information from local and regional data sources, and substantially increased the number of, and level of detail about, GLOFs in existing compilations. Compared to the global survey by Carrivick and Tweed (2016), the number of reported



GLOFs has increased from 1,348 to 3,151 cases (+134%). We could add 240 new GLOFs to our inventory alone since the study by Carrivick and Tweed (2016). By truncating both databases to the same time interval (1500-2015), we find that NW

North America had the highest increase in the absolute number of reported GLOFs with 319 (+88%) additional cases (**Fig. 10**). Greenland had only 22 GLOFs in the previous global database, and our inventory raises the GLOF count there 6-fold, followed by High Mountain Asia (+155%) and Iceland (+92%). In the Andes, the European Alps, Scandinavia, and Greenland, the increase in the number of GLOFs is evenly distributed with time (**Fig. 8**). In contrast, most newly added events in NW North America, High Mountain Asia, and Iceland cover the period after 1950. The strong increase in the reported GLOFs in

only six years following the study of Carrivick and Tweed (2016) might mirror the increase in research interest in this glacial hazard and better documentation. In High Mountain Asia and Greenland, the increase is mainly attributable to recently published reviews by Bhambri et al. (2019), Carrivick and Tweed (2019), and Bazai et al. (2021). These studies systematically compiled all accessible information on previously reported cases, while reporting only a few newly detected GLOFs. In NW North America and Scandinavia, publicly available online databases have offered better access to data (NVE Glacier inventory,

http://glacier.nve.no/Glacier/viewer/GLOF/en/, last accessed 02 November 2022; Alaska Glacier Dammed Lake Database, https://www.weather.gov/aprfc/gdlMain, last accessed 19 October 2022). In Iceland, the increase in reported GLOFs is due local experts, grating us access to internal reports from the government agencies and translated publications written in Icelandic language. Using this newly accessible information, we were able to add 321 GLOFs to that region. Among those, 251 cases occurred before 2016, accounting for 43% of the GLOFs that have been reported in Iceland.

Moreover, the substantial increase in GLOF reports since the study by Carrivick and Tweed (2016) highlights the advances in data collection. Many recent publications and web resources used remote sensing to detect GLOFs. For example, Emmer et al. (2022b) used multi-temporal aerial and satellite imagery, obtained between 1948 and 2022, to detect and analyse GLOFs that occurred in Peru and Bolivia and tripled the number of previously reported events in these regions. Veh et al. (2019) doubled the known amount of moraine-dam failures in the Himalayas between 1988 and 2017 by evaluating time series of

Landsat images. Published hydrometric data also improved reporting in sparsely populated regions. The Alaska Glacier Dammed Lake Database (https://www.weather.gov/aprfc/gdlMain, last accessed 19 October 2022) reported 83 GLOFs for the period after 2015 alone, mainly based on USGS hydrometric station records. These data account for 81% of the reported GLOFs in NW North America since 2015.





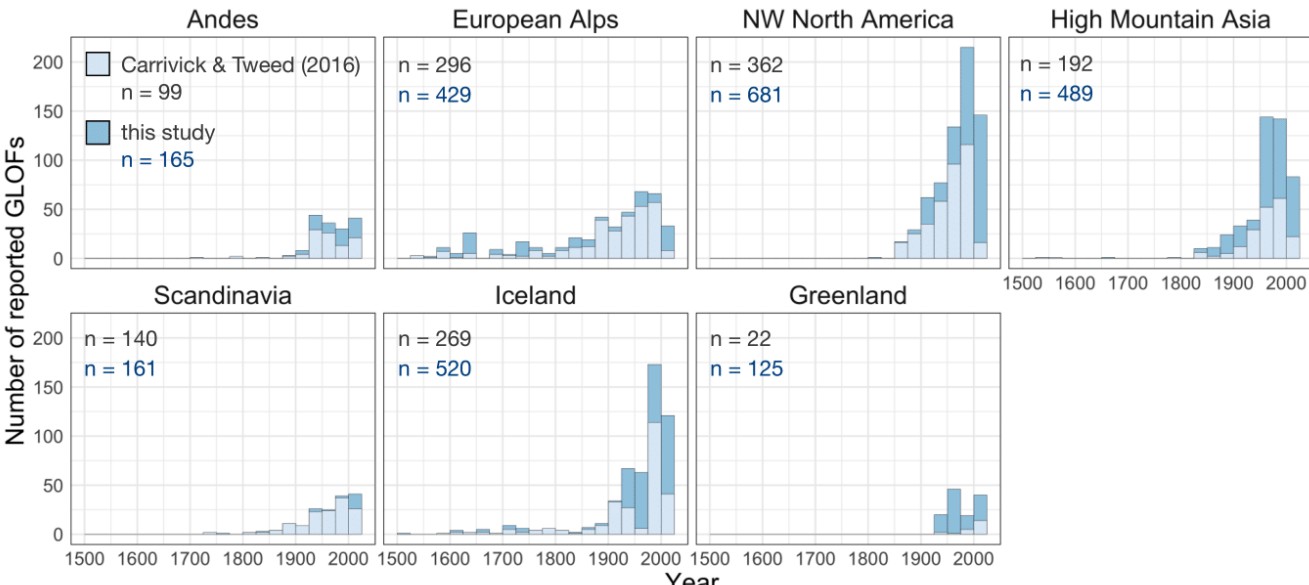

**Figure 10: Number of reported GLOFs with yearly timestamps in 25-year bins.** We compare data from Carrivick and
Tweed (2016) and our study between 1500 and 2015, and exclude five GLOFs that happened before 1500 for display purposes.

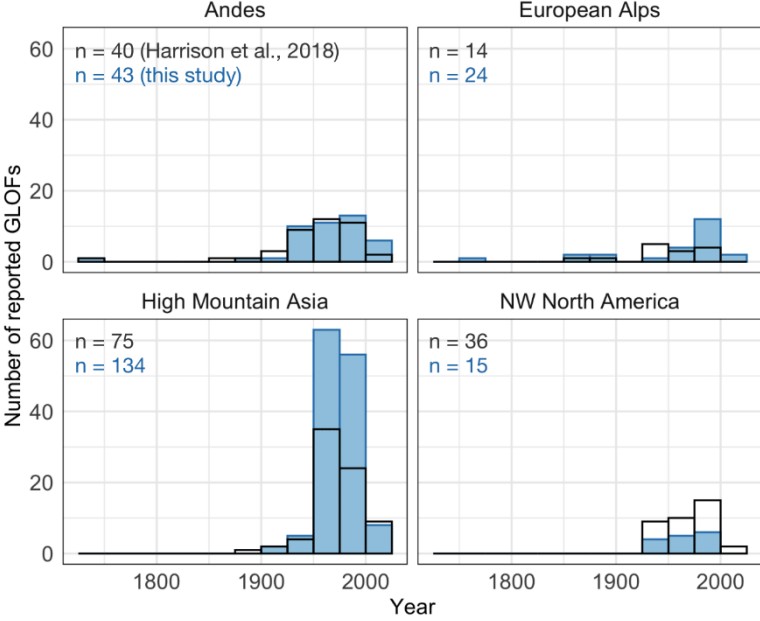

**Figure 11: Number of reported outbursts from moraine-dammed lakes with explicit year reported by Harrison et al.
(2018) and this study.** Data are aggregated into 25-year bins and refer to the same regions and period (1725-2011).

We filtered our database for moraine-dammed lakes and compared them to the study by Harrison et al. (2018) during the same
observation period (1725-2011). We find that our database contains 51 additional cases (+31%), but fewer cases in NW North



America (**Fig. 11**). We recall that our main criteria for data selection is accurate information on location, timing, and a reference. Hence, we discarded all cases with vague descriptions of their origin (i.e. only country or mountain range), without a distinct timestamp or time range of occurrence, or without a reference. For example, we excluded two GLOF from moraine-dammed lakes in British Columbia, in 1965, and Pakistan, in 1878, due to the lack of further locational information or any
further description of the flood.

## 4.2 Limitations and possibilities for improvement

Our inventory shows that most GLOFs were reported with some delay, likely causing the decrease in the annual number of reported GLOFs between 2020 and 2022 (**Fig. 6**). Immediately reported GLOFs such as those determined by hydrometric stations, field photographs and videos, or satellites with high repeat rates are usually published through governmental websites
or social media. These platforms help detect recent changes in GLOF activity and magnitude, and have the potential to inform early warning systems (Huggel et al., 2020; Wang et al., 2022). Most social media posts in our database were published either by public institutions (e.g. UNDP Pakistan, https://mobile.twitter.com/undp_pakistan, last accessed: 14 November 2022) or researchers and therefore can be considered as being reliable. In a few cases, posts on social networks mistook mass flows such as debris flows, flash floods, or rock ice-avalanches for GLOFs. Future validation processes could be eased and improved
using remote sensing products offering high spatial and temporal resolution. However, at present, access to these products is often restricted to specific user groups or to commercial licenses. In such cases, local experts are important for identifying and validating individual GLOFs and maintaining quality standards of our database. We acknowledge that our database misses an unknown number of GLOFs because these cases are neither reported nor are the underlying data freely accessible. Consequently, we thoroughly analysed possible reporting biases on an earlier version (V2.0) of our inventory (Veh et al.,
2022). Our estimates indicate that in the early to mid-20th century on average two to four out of five GLOFs might have gone unnoticed. Since the 1970s, remote sensing has improved GLOF detection and reporting, in particular with the launch of continuous satellite missions such as the Landsat program (Wulder et al., 2022).

We further acknowledge that some of our numeric parameters such as $V_L$, $V_0$, and $Q_p$ come with uncertainties since reporting standards might not be consistent throughout the database. Stream gauges sample GLOF discharges at different distances from
the source, as flood data is often obtained close to build-up areas (Krabbenhoft et al., 2022). Gauged or estimated values of $Q_p$ may or may not include the baseflow of the affected river (Carrivick and Tweed, 2016). Moreover, installing stream gauges largely depends on the accessibility and the local river properties, making a standardised gauging scheme less practical on global scale. We accounted for these differences in measuring and reporting by compiling all available information on how values were obtained in our inventory. However, some data services publish information on GLOF peak discharge and volume
without information on the underlying methods, and might need further quality control. Regional inventories offer extensive collections of flood magnitudes from previous studies, but hardly disclose how those measurements were obtained (Alaska Glacier Dammed Lake Database, https://www.weather.gov/aprfc/gdlMain, last accessed 19 October 2022; Bhambri et al., 2019; Bazai et al., 2021). Hence, the method for estimating flood volume or the technical equipment for measuring flood



discharges remains unknown in some cases. We recommend that future studies will be explicit on how values were obtained
given that $V_0$ and $Q_p$ became the most commonly used parameters to describe GLOF magnitudes (Walder and Costa, 1996;
Harrison et al., 2018).

Many of the missing parameter values might be explained by the lower emphasis and less detailed description in former studies.
For example, damage data might be less complete in some regions because governmental responsibilities and political priorities
are unclear or difficult to follow (Carrivick and Tweed, 2016). Some studies simply put GLOFs in a broader context, such as
in describing regional geomorphology or hazards, with less focus on the flood characteristics of single outbursts (Brabets,
1996; Fort, 2015). Consequently, some of the included parameters leave room for improvement with regards to data density.
For example, the amount of cases with reported lake volume $V_L$ and flood volume $V_0$ could potentially be improved by using
our mapped lake area data, for example by intersecting the lake outline with bathymetric data (Muñoz et al., 2020). Estimates
of $V_L$ and $V_0$ might also come from empirical relationships between lake area and volume (Cook and Quincey, 2015; O'Connor
et al., 2001) or geometric lake parameters such as the lake width to length ratio (Qi et al., 2022) because the lake bathymetry
is often unknown. Similarly, data density of $Q_p$ could potentially be improved by using scaling relationships between $Q_p$ and
$V_0$ (Clague and Mathews, 1973; Evans, 1986), or the diameter of transported boulders (Strand, 1977; Costa, 1983; Gurung et
al., 2017). Radar data with high spatial resolution and comparably short revisit times, such as Sentinel-1
(https://sentinels.copernicus.eu/web/sentinel/missions/sentinel-1, last accessed 14 November 2022), could be used to improve
the data density of $A_b$ and $A_a$. Radar is particularly useful in regions with frequent cloud cover such as Iceland because the
signal can penetrate clouds (Wangchuk and Bolch, 2020).

Following our parameter scheme, any newly available information on historic and recent GLOFs can now be categorized and
implemented in our inventory. Our database can be extended by parameters that may become necessary for future analysis to
assess links between GLOF triggers, glacier decay, and GLOF activity and magnitude. For example, the glacier area derived
from the RGI provides important insights into local glacier settings, but shows only a snapshot of the ice extent. This issue
became particularly visible at some former GLOF locations at which the local glacier had already disappeared until the time
the RGI was created. Thus, the RGI data are less suitable to quantify and link local glacier changes with the reported GLOFs
especially given accelerating rates of glacier retreat in recent years (Hugonnet et al., 2021). Therefore, parameters describing
the dynamic glacier properties, for instance the surge or retreat rate, might be more useful than the glacier area and thus could
be included in future versions of our database. Additional parameters might also be added to better differentiate between GLOF
triggers. New parameters might address if a GLOF followed extreme weather events, such as exceptionally high precipitation,
or if the outburst is likely or known to be triggered by other external sources, for example avalanches or landslides into the
lake. For now, any information about the cause of a GLOF are only stored in the comments section. Parameters describing the
local river system could further provide a base for the assessment of downstream flood impact. Informative parameters might
include the length or width of the stream, and the type and amount of transported material. However, these attributes might be
prone to change since some of the floods have major geomorphic impacts and reshape the river. We further aim to extend our

global inventory by focusing on regions beyond our main study regions (**Fig. 1**), for example remote regions such as NE Canada, Antarctica, and Arctic Russia.

## 5 Data availability

The global GLOF database (V3.0) is archived at https://doi.org/10.5281/zenodo.7330345 (Lützow and Veh, 2022). The data are split by regions into separate tables in an OpenDocument Spreadsheet that can be opened with free software such as LibreOffice or Apache OpenOffice. In each table, the columns denote the parameters, and rows are unique GLOFs.

Our database is an ongoing project, and we offer a web-based, interactive map that grants access to the most recent state of the database (http://glofs.geoecology.uni-potsdam.de). Users can also download all previous versions of the database from this

interface.

The initial version (V1.0) of the database was released on 2021-06-10, and included information on GLOF location, timing, outburst characteristics, and references. Data on manually mapped lake areas were added with Version 2.0 (2022-03-01). The current version V3.0 (2022-11-17) offers categorized data on damages.

## 6 Conclusions

We collated a global GLOF inventory that more than doubles the number from previous appraisals. We propose a standardised protocol for reporting characteristic GLOF diagnostics such as location, date, hydrodynamic flood characteristics, flood impact, and reference. Following this approach, our collated database allows for objective comparisons on different spatial or temporal scales. Potential analysis based on the data might concern trends in GLOF occurrence, magnitude and impact, providing a valuable base for future hazard, risk assessment, and early warning. We find regional differences in GLOF

reporting, reflected in the type and amount of available research items. Areas with comparably high GLOF activity are potentially attributable to reporting biases. In this context, we acknowledge gaps in our database. Some of these gaps could be filled using the current parameters in combination with additional datasets. This study strongly benefited from the work with local experts, highlighting the importance of joint contribution within the research community. This motivates a versioned, continuously updated inventory, ultimately required to improve flood impact mitigation, including new strategies for

sustainable socio-economic development and the management of the fragile ecosystems that get temporarily disturbed by these floods.

**Author contributions.** NL and GV designed the study and collected data of historic GLOFs. NL led the writing of the manuscript, produced all figures, curated the database, and maintains the web interface. All authors contributed in writing the manuscript.

**Competing interests.** GV is a topical editor at Earth System Science Data.





**Acknowledgements.** We are indebted to many colleagues who have helped collect data and shared their knowledge on historic outbursts: Pablo Iribarren Anacona; Jonathan Carrivick; Bergur Einarsson; Adam Emmer; Marten Geertsema; Matthias Huss; Miriam Jackson; Rex Johnston; Varvara Kharlamova; Dmitry Petrakov; Vanessa Round; Jakob Steiner; Fiona Tweed; and Mauro Werder. We thank Jenny Tamm for mapping the outlines of glacier lakes. We acknowledge a basic license from the Education and Research program from Planet Labs Inc. through the PlanetExplorer (https://www.planet.com/) and free access to the Landsat archive via the USGS Earth Explorer (https://earthexplorer.usgs.gov/) to map glacier lakes.


**Financial support.** This research has been supported by the German Research Foundation (Deutsche Forschungsgemeinschaft DFG) under grant number VE 1363/2-1.



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
