# Peer review of "A global database of historic glacier lake outburst floods"

_Earth System Science Data, 2022_

## Author Comment (AC1)

We thank the reviewer for their comprehensive and constructive comments on our work. Below, we respond to their comments in blue font and describe how we will address these comments in the revised manuscript in black font. References to specific lines refer to the initial manuscript.

**#Referee 1**

**R1C1:** It is always a pleasure to review a well-written manuscript presenting interesting data such as this one. The authors compiled unprecedently detailed inventory of GLOFs that happened on Earth in past ca 1300 years. This dataset has potential to be cited in studies dealing with GLOF occurrence patterns in space and time and GLOF hazard and risk assessment studies, and I find it generally suitable for ESSD. I have two general and few specific comments (below) to this manuscript:

**R1A1:** We thank the reviewer for recognising the value and potential applications of our database, and address any remaining comments in our point-by-point replies below.

**General Comments:**

**R1C2:** While this is a data science paper and the authors declare that "the goal is to present a global dataset that covers consistently and systematically the largest number of reported GLOFs" (L58-59), only limited methodological details are actually revealed in Section 2.1. I wonder what was the workflow of systematic search for GLOFs (i) in peer-reviewed literature; and especially (ii) in non-peer reviewed literature, news and social media. I acknowledge that it is nearly impossible to go through all diverse sources but I feel that the authors should make it clear what types of sources (news outlets, social media, repositories of local authorities, DRR reports, …) they went through and how. What about non-English sources? I find this especially important because it will allow future researchers easily identify and address possible gaps and avoid work redundancy. So please provide more details on your GLOF search methodology and its consistency and systematic aspects.

**R1A2:** We agree with points raised by the reviewer. In our revised manuscript, we will extend the section on literature search as follows (L71): "*We compiled these sources by searching online literature archives (e.g., Web of Science), and web platforms that are commonly used by researchers, governmental agencies, or newspapers to spread news and new research (e.g., ResearchGate, Google Scholar, Twitter). We have repeatedly revisited these databases over a four-year period (2019-2022) to include any updates and new literature in our inventory. Guided by previous literature reviews (Emmer et al., 2018, 2022), we used a keyword search on these platforms including terms in English language such as "glacial lake", "glacial lake outburst flood", "glacier floods", "glacier hazards", and added local or regional names such as "Himalaya", "Canada" or "Patagonia" to obtain more specific research items. We traced back each reported GLOF to the original source, and contacted the authors or consulted libraries if the sources were not available online.*" Those cases were then checked in more detail (L84): "*All resources were comprehensively checked for any available information on GLOF events (…) We compiled 57 attributes to capture quantitative and qualitative information on location; timing; hydrodynamic characteristics; socio-economic and geomorphic impacts; and references associated with all reported GLOF in our database (**Fig. 3**). We summarize these attributes below.*"

We acknowledge that our search was guided by a stepwise, case-by-case selection over several years. Other scientific disciplines have used "systematic" reviews to search for one or more specific keys and downloading the entire literature archive at a given point in time. Such a compilation, though desirable, is beyond reach: as of April 28, 2023, Google Scholar returns almost 1 million results for the key "glacial lake".

**Screenshot of Google Scholar search for the key word "glacial lake" as of April 28, 2023**.

We will therefore remove the term "systematic" in our manuscript and change it to more appropriate wording to remain consistent with previous appraisals.

We will further add information on the underlying languages of our resources (L75): *"We have compiled GLOFs from literature sources written in English, Russian, German, Spanish, Icelandic, and Chinese. Sources not written entirely in English must include at least an abstract and keywords in English to meet our search criteria. We were also supported by 14 local researchers who reviewed our compilation and contributed additional cases with their local knowledge (see Acknowledgements). With their help, we were able to expand the previously available catalogue of GLOFs substantially, especially in Iceland and Central Asia (Carrivick and Tweed, 2016).*

**R1C3:** The authors refer to different versions of their database. One (let's call it 'static' database) is published at Zenodo (allright), while the other ('active') is found at Uni Potsdam website. I'm convinced that future development of the active one is especially important and interesting, because new GLOFs will happen / unreported ones will be discovered. Yet, it is not clear how newly discovered or recently occurred GLOFs can be added (?), following the statement of "joint contribution within the research community". This goes hand in hand with what you mentioned on L316-317, i.e. "local experts are important for identifying and validating individual GLOFs.". Absolutely, yes (and this is something we called for already in our 2016 effort, and we failed; 10.1007/s10346-016-0686-6), but what is your strategy of ensuring that? Please make clear how can a person contribute to this joint effort and what is the future (and sustainability) of this 'active' database branch on the one hand; and how will you ensure quality standards among various contributors on the other hand?

**R1A3:** We agree that active input from the research community is key to successfully maintaining our database and avoiding major data gaps. We followed the reviewer's idea and implemented a submission form on our website http://glofs.geoecology.uni-potsdam.de/ to ease the reporting of missing events for other researchers (see screenshot below). We will review each submitted case or reference individually and add these new observations to the next version release of the database. To inform the reader about this option, we will add to the data availability section (L373): *"Our database is an ongoing project, and we offer a web-based, interactive map that grants access to the most recent state of the database (http://glofs.geoecology.uni-potsdam.de). (…) This website includes a submission form that enables the user to report missing or recently occurred GLOFs (**Fig. 12**)"*. Considering future updates, Zenodo will remain the version-controlled access to our database. Zenodo allows us to update our database under a new version using the same DOI, and we are going to release a version 3.1 as soon as a sufficient amount of new data has gathered (see also our reply **R2A2** to reviewer #2). Furthermore, we plan to discuss schemes for standardized reporting and joint contribution within the GLOF community at an upcoming GSA Penrose conference on outburst floods (https://www.jsg.utexas.edu/penrose-2023/).

[Figure]

**Figure 12: Submission form on** http://glofs.geoecology.uni-potsdam.de **that users can use to report missing or recently occurred GLOFs.**

**Specific comments:**

**R1C4**: L10: magnitude, frequency and impacts of processes …

**R1A4:** We will change our wording accordingly.

**R1C5**: L45: is there any specific reason for mentioning this particular GLOF?

**R1A5:** We mention this GLOF as an example of the historic catastrophic consequences of GLOFs. There was no specific reason for mentioning this particular event, and welcome any other suggestions to emphasize the societal impacts from GLOFs.

**R1C6:** L62: more info about the workflow and explored types of resources is needed (see my general comment)

**R1A6:** We refer the reviewer to our reply R1A2.

**R1C7:** L96: strictly speaking, lake dam type is not a characteristic of a location

**R1A7:** We agree and will relabel the box in Figure 3 to "Location and lake characteristics".

**R1C8:** L129: did you map lakes or GLOF (impacts)?

**R1A8:** We will clarify the sentence as follows (L129): *"We manually mapped the outlines of lakes that burst out between 1984 and 2021 using multispectral optical images from eight satellite missions:*

*Landsat 4, 5 (TM), 7 (ETM+), and 8 (OLI), Geo-Eye 1, RapidEye, Sentinel-2, and Planetscope.*  *(…) In each case, we selected images as temporally close as possible before and after the outburst date, filtering images with cloud cover of <20%, and minimum snow and shadow cover."*

**R1C9:** L129-136: this procedure applies also for Dates, right?

**R1A9:** Yes; we will add the following paragraph to the date section to highlight this (L109):

*"For cases without reported date, we used optical satellite imagery (e.g., Landsat, RapidEye, and Planetscope) to determine a range of plausible dates. We further used this approach to narrow reported date ranges when feasible."*

**R1C10:** L221: I guess that this recent gap 2021-2022 is rather due to the reporting lag (I'm aware of at least one GLOF from moraine-dammed lake in 2022: https://repositorio.inaigem.gob.pe/items/c0beb0b1-e989-41e4-a785-8d5b699a48de) that leads me to my general comment about systematic search among various resources and the need to specify that in more detail)

**R1A10:** The thank the reviewer for bringing this recent case to our attention, and will include it our next version of our database. We discuss limitations and reasons for data gaps in section 4.2 (see L307): *"Our inventory shows that most GLOFs were reported with some delay, likely causing the decrease in the annual number of reported GLOFs between 2020 and 2022 (**Fig. 6**). Immediately reported GLOFs such as those determined by hydrometric stations, field photographs and videos, or satellites with high repeat rates are usually published through governmental websites or social media."*

**R1C11:** Fig. 7: intuitively, I would expect no data in grey color

**R1A11:** We thank the reviewer for pointing this out. We will reverse the colour scale in Fig. 7 accordingly**:**

[Figure]

**Figure 7 with adjusted colour scale.**

**R1C12:** L242: those % should be related to share of individual lake dam types among GLOF-producing lakes

**R1A12:** We are unsure about this comment. We feel that these percentages show that ice-dammed lakes are abundant and also frequently caused damage.

**R1C13:** L353-368: please consider a separate sub-section discussing the future of the active branch of your database and ways how local experts are planned to be involved (see my general comment)

**R1A13:** We refer the reviewer to our previous comment R1A3.

**R1C14:** L368: Interesting, would you consider outbursts of thermokarst lakes in high latitude regions to be GLOFs? That would increase a total numbers a lot (see e.g. http://doi.org/10.1002/ppp.2038)

**R1A14:** We exclude thermokarst lakes as GLOF sources from our database. Thermokarst lakes are not necessarily fed by meltwater from glaciers and are therefore different from the formal definition of a glacier lake (Carrivick and Tweed, 2013).

**R1C15:** L369: please make clear what should preferably be cited if your data are used – zenodo repository, uni-potsdam website or this paper?

**R1A15**: The goal of articles considered in ESSD is to *"enable the reader to review and use the data, respectively, with the least amount of effort. To this end, all necessary information should be presented through the article text and references in a concise manner and each article should publish as much data as possible"* (https://www.earth-system-science-data.net/about/aims_and_scope.html). The data itself can be archived in any citable and freely accessible source (i.e. including a digital object identifier, DOI, as Zenodo does). We therefore would like to leave this choice to the users whether they cite the data description paper or the database itself.

**R1C16:** To sum up, I support the publication of this manuscript once some moderate revisions addressing my two general comments are made.

**R1A16:** We highly appreciate the reviewers support for publication of our work and are confident that we will be able to address all general comments in a revised manuscript.

**References:**

Carrivick, J. L. and Tweed, F. S.: Proglacial Lakes: character, behaviour and geological importance, Quaternary Science Reviews, 78, 34–52, 2013.

Emmer, A.: GLOFs in the WOS: bibliometrics, geographies and global trends of research on glacial lake outburst floods (Web of Science, 1979–2016), Nat. Hazards Earth Syst. Sci., 18, 813–827, https://doi.org/10.5194/nhess-18-813-2018, 2018.

Emmer, A., Allen, S. K., Carey, M., Frey, H., Huggel, C., Korup, O., Mergili, M., Sattar, A., Veh, G., Chen, T. Y., Cook, S. J., Correas-Gonzalez, M., Das, S., Diaz Moreno, A., Drenkhan, F., Fischer, M., Immerzeel, W. W., Izagirre, E., Joshi, R. C., Kougkoulos, I., Kuyakanon Knapp, R., Li, D., Majeed, U., Matti, S., Moulton, H., Nick, F., Piroton, V., Rashid, I., Reza, M., Ribeiro de Figueiredo, A., Riveros, C., Shrestha, F., Shrestha, M., Steiner, J., Walker-Crawford, N., Wood, J. L., and Yde, J. C.: Progress and challenges in glacial lake outburst flood research (2017–2021): a research community perspective, Nat. Hazards Earth Syst. Sci., 22, 3041–3061, https://doi.org/10.5194/nhess-22-3041-2022, 2022.

---

## Author Comment (AC2)

We thank the reviewer for their comprehensive and constructive comments on our work. Below, we respond to their comments in blue font and describe how we will address these comments in the revised manuscript in black font. References to specific lines refer to the initial manuscript.

**#Referee 2**

**R2C1:** The paper describes a new global inventory on GLOFs that is claimed to more than double the number of reported GLOFs in a previous global inventory.

The topic is extremely acute as global deglaciation has brought about skyrocketing number of new glacial lakes and increase in potential danger.

**R2A1:** We thank the reviewer for highlighting the relevance of our work.

**General Comments:**

**R2C2:** Brief examination on such underreported regions as Caucasus, Tajikistan, Kyrgyzstan, Afghanistan shows that authors put real effort in registering as many GLOFs as it is possible. But still some of known cases for Caucasus and Central Asia are not presented in the database because they were reported in Russian language publications. Just a brief example: more than 30 GLOF locations in Kyrgyzstan is reported here: http://ru.mes.kg/Kniga/book_rus078.html

*Мониторинг, прогнозирование опасных процессов и явлений на территории Кыргызской Республики* (Изд. 18-е с изм. и доп.), Б.: МЧС КР, 2021 - 819 с.

Monitoring, forecast of dangerous processes and phenomena in Kyrgyzstan Republic (18th Edition). Bishkek: MCHS KR, 2021 – 819 p. (in Russian)

While in the presented inventory includes 17 locations in Kyrgyzstan.

That is probably out of the scope of the paper to work with sources in local languages, but still this problem and potential perspective for development needs to be mentioned and discussed.

Some of additional cases for the Caucasus can be found here:
https://link.springer.com/article/10.1134/S009780782207003X

**R2A2:** We thank the reviewer for pointing out the missing cases in the Caucasus and Central Asia. We contacted local native speakers to help us adding these GLOFs to our database. We are currently identifying the exact location of the source lakes, as they are largely mentioned by their local names, rather than by coordinates. We would be thankful if the reviewer could provide us more detail on the source coordinates of these lakes, for example by using the submission form on our website that we have recently added (see our reply **R1A3** to reviewer #1). Those cases will be archived soon under a new version (3.1) on the same DOI on Zenodo.

We will also add information on underlying languages of our references to the method section. We would like to refer the reviewer to our reply **R1A2**, which we copy here for convenience:

**R1A2:** We will further add information on the underlying languages of our resources (L75): *"We have compiled GLOFs from literature sources written in English, Russian, German, Spanish, Icelandic, and Chinese. Sources not written entirely in English must include at least an abstract and keywords in English to meet our search criteria. We were also supported by 14 local researchers who reviewed our compilation and contributed additional cases with their local knowledge (see Acknowledgements).*

*With their help, we were able to substantially expand the previously available catalogue of GLOFs, especially in Iceland and Central Asia (Carrivick and Tweed, 2016)."*

**Specific Comments:**

The paper overall is well written and well-illustrated, anyway there are still some points that need improvement or correction:

**R2C3:** Not all study regions are plotted on Fig.1

**R2A3:** We will adjust the figure accordingly:

[Figure]

**Figure 1 with added study region 'Other' (New Zealand/ Caucasus).**

**R2C4:** It is not clear if data on area before (Ab) and after the GLOF (Aa) was reported in the literature was it included in the database. Or all values in the database are based on performed analysis based on satellite imagery.

**R2A4:** We manually mapped the areas before and after the GLOF (Ab and Aa) from satellite images unless stated otherwise in our manuscript (L143-144): *"We also included in our database the lake outlines from 11 GLOFs mapped by Bazai et al. (2021) in the Karakoram, and nine GLOFs mapped by Eide (2021) in Scandinavia."*

**R2C5:** The database would benefit from adding mapped glacial lakes outlines before and after the GLOF

**R2A5:** We are currently compiling more data on lake areas before and after the outburst to foster a more complete dataset. We will publish the lake polygons before and after the GLOF on the current DOI on Zenodo once this is achieved.

**R2C6:** The authors mention source types in Methods section of the paper, but there is no such field in the database. Including it would benefit the database.

**R2A6:** We thank the reviewer for pointing this out. We categorized the single sources in our citation manager. Users can derive the underlying source type from citations in the reference column. Resources cited in other databases, including their language (if not English), are indicated in this column (L166-170): *"Finally, we listed all sources from which we extracted information on GLOFs. We highlighted earlier published information that was cited in more recent publications by linking them with "CITED IN", independent of the accessibility to the cited source. If we had access to a cited reference, we always searched for the original source to validate the provided information. If a publication provided multiple sources for an event (e.g. in data tables), the cited references were connected with an "&" operator."*

**R2C7:** For some regions (for ex. Caucasus) approach to sorting the event is not clear (not date of the event). Please check that.

**R2A7:** We thank the reviewer for pointing this out. We will adjust the order accordingly and upload the updated database file on the Zenodo repository (see screenshot below). In the study region 'Other', which includes GLOFs from two spatially separated regions, we decided to sort the entries first by country (i.e., Caucasus, New Zealand), then by Date, to maintain an order consistent with the other sheets.

| ID | Major_RGI | Mountain_r | Country | Glacier | RGI_Glacier | RGI_Glacier | Lake | Lake_type | Longitude | Latitude | River | Date | Date_Min | Date_Max | Mecha |
|---|---|---|---|---|---|---|---|---|---|---|---|---|---|---|---|
| running | RGI region in | Major | Source | Name of the | Glacier Id | Glacier area km² | Name of the | e.g ice, | X coordinate XX.XX° | Y coordinate XX.XX° | Major river | Reported YYYY-MM- | Earliest YYYY-MM- | Latest YYYY-MM- | (e.g. |
| | | e.g. | e.g. Pakistan | e.g. Baltoro | | | e.g. Baltoro | | | | e.g. Indus | | | | |
| 1 | Caucasus, | Greater | Russia | Birdzhalychir | RGI60-12.01 | 20,98 | | unknown | 42,530 | 43,394 | Malka | 1909-08-02 | | | unknow |
| 5 | Caucasus, | Greater | Russia | Bashkara | RGI60-12.00 | 3,66 | Bashkara | moraine | 42,725 | 43,209 | Baksan, | 1958 | | | overtop |
| 6 | Caucasus, | Greater | Russia | Bashkara | RGI60-12.00 | 3,66 | Bashkara | moraine | 42,725 | 43,209 | Baksan, | 1959 | | | overtop |
| 12 | Caucasus, | Greater | Russia | unknown | | | Kakhab- | water pocket | 46,554 | 42,220 | Sulak | 1969-08-28 | | | englaci |
| 13 | Caucasus, | Greater | Russia | unknown | | | Kakhab- | water pocket | 46,554 | 42,220 | Sulak | 1971-07 | 1971-07-01 | 1971-07-31 | englaci |
| 14 | Caucasus, | Greater | Russia | unknown | | | Kakhab- | water pocket | 46,554 | 42,220 | Sulak | 1974-08-02 | | | englaci |
| 15 | Caucasus, | Greater | Russia | unknown | | | Kakhab- | water pocket | 46,554 | 42,220 | Sulak | 1974-08-06 | | | englaci |
| 16 | Caucasus, | Greater | Russia | unknown | | | Kakhab- | water pocket | 46,554 | 42,220 | Sulak | 1975-07-14 | | | englaci |
| 17 | Caucasus, | Greater | Russia | unknown | | | Kakhab- | water pocket | 46,554 | 42,220 | Sulak | 1975-07-24 | | | englaci |
| 8 | Caucasus, | Greater | Russia | Malyi Azau | RGI60-12.00 | 11,81 | Malyi Azau | moraine | 42,447 | 43,284 | Baksan, | 1978-07-19 | | | overtop |
| 2 | Caucasus, | Greater | Russia | Birdzhalychir | RGI60-12.01 | 20,98 | | moraine | 42,531 | 43,372 | Malka | 1993 | 1993-06-01 | 1993-10-31 | overtop |
| 3 | Caucasus, | Greater | Russia | Birdzhalychir | RGI60-12.01 | 20,98 | | ice | 42,531 | 43,372 | Malka | 2003 | 2003-06-01 | 2003-11-24 | overtop |
| 4 | Caucasus, | Greater | Russia | Birdzhalychir | RGI60-12.01 | 20,98 | | ice | 42,531 | 43,372 | Malka | 2006-08-11 | | | overtop |
| 10 | Caucasus, | Greater | Russia | Rakyt | | | | water pocket | 43,159 | 43,180 | Chegem, | 2007-08-02 | | | englaci |
| 11 | Caucasus, | Greater | Russia | Passionaria | RGI60-12.01 | 0,149 | | water pocket | 43,903 | 42,764 | Ardon, Terek | 2007-08-02 | | | englaci |
| 9 | Caucasus, | Greater | Russia | Malyi Azau | RGI60-12.00 | 11,81 | Malyi Azau | moraine | 42,447 | 43,284 | Baksan, | 2011-11-08 | | | overtop |
| 7 | Caucasus, | Greater | Russia | Bashkara | RGI60-12.00 | 3,66 | Bashkara | water pocket | 42,725 | 43,209 | Baksan, | 2017-09-01 | | | overtop |
| 18 | New Zealand | Southern | New Zealand | Franz Josef | RGI60-18.02 | 33,11 | | ice | 170,172 | -43,443 | Waiho | | 1920 | 1940 | breach |
| 19 | New Zealand | Southern | New Zealand | Franz Josef | RGI60-18.02 | 33,11 | | unknown | 170,172 | -43,443 | Waiho | 1949-02 | | | unknow |
| 20 | New Zealand | New Zealand | New Zealand | Mangaturutu | RGI60-18.00 | 1,18 | Mount | ice/sediment | 175,564 | -39,281 | Whangaehu | 1953-12-24 | | | breach |
| 21 | New Zealand | Southern | New Zealand | Franz Josef | RGI60-18.02 | 33,11 | | unknown | 170,172 | -43,443 | Waiho | 1965-12-19 | | | unknow |
| 22 | New Zealand | Southern | New Zealand | Franz Josef | RGI60-18.02 | 33,11 | | ice | 170,172 | -43,443 | Waiho | 1967-01 | | | breach |
| 23 | New Zealand | Southern | New Zealand | Franz Josef | RGI60-18.02 | 33,11 | | water pocket | 170,172 | -43,443 | Waiho | 1967-03 | | | sub-/ |
| 24 | New Zealand | Southern | New Zealand | Franz Josef | RGI60-18.02 | 33,11 | | supraglacial | 170,172 | -43,443 | Waiho | 1981-06-02 | | | overtop |
| 25 | New Zealand | Southern | New Zealand | Maud Glacier | RGI60-18.02 | 10,336 | unknown | moraine | 170,500 | -43,476 | Godley | 1992-05-03 | | | unknow |
| 26 | New Zealand | Southern | New Zealand | Maud Glacier | RGI60-18.02 | 10,336 | unknown | moraine | 170,500 | -43,476 | Godley | 1992-09-16 | | | unknow |
| 27 | New Zealand | Southern | New Zealand | Franz Josef | RGI60-18.02 | 33,11 | | water pocket | 170,172 | -43,443 | Waiho | 1993-09 | | | sub-/ |
| 28 | New Zealand | Southern | New Zealand | Franz Josef | RGI60-18.02 | 33,11 | | unknown | 170,172 | -43,443 | Waiho | 1994-01 | | | unknow |
| 29 | New Zealand | Southern | New Zealand | Franz Josef | RGI60-18.02 | 33,11 | | water pocket | 170,172 | -43,443 | Waiho | 1995-12-13 | | | sub-/ |
| 30 | New Zealand | Southern | New Zealand | Franz Josef | RGI60-18.02 | 33,11 | | water pocket | 170,172 | -43,443 | Waiho | 1997-05 | | | sub-/ |
| 31 | New Zealand | Southern | New Zealand | Franz Josef | RGI60-18.02 | 33,11 | | water pocket | 170,172 | -43,443 | Waiho | 1998-02 | | | sub-/ |
| 32 | New Zealand | Southern | New Zealand | Franz Josef | RGI60-18.02 | 33,11 | | unknown | 170,172 | -43,443 | Waiho | 1998-03 | | | unknow |
| 33 | New Zealand | Southern | New Zealand | Franz Josef | RGI60-18.02 | 33,11 | | supraglacial | 170,172 | -43,443 | Waiho | 2003-02-14 | 2003-02-14 | 2003-03-05 | unknow |
| 34 | New Zealand | Southern | New Zealand | Fox Glacier | RGI60-18.02 | 34,72 | | supraglacial | 170,087 | -43,501 | Fox | 2007-01-12 | | | unknow |

**Screenshot of adjusted database file.**

**R2C8:** Fields from reported_impacts to reported_fatalities include letters (u/x/a) and figures. It needs to be transcribed in the text. It is also not the best idea to use both character and numeric data in one filed.

**R2A8:** We will add this information in L159: *"If available, we also documented the number of damaged features of each category. Where information on GLOF impacts was vague, we distinguished between features that were damaged (x) or only affected (a) without structural damage, for example by covering a road with debris. When damage was reported without information on the affected features, the cases are marked with a 'u' (unknown) in the reported impacts parameter."*

**R2C9:** What is a location identifier in the base?

**R2A9:** We are unsure which topic or text passage this comment refers to.

**R2C10:** It would be useful to have information on total number of fatalities, destroyed buildings etc. (globally and regionally)

**R2A10:** We will add to our manuscripts (L241): *"According to our database, 44 GLOFs have killed at least 3,636 people. Most fatalities (n = 3,093) were reported in HMA; Iceland only had few (n = 7) and Scandinavia and Greenland had no reported fatalities. We note that quantifying the absolute amount of damage and the absolute number of fatalities from individual GLOFs can be prone to both over- and underestimations. For example, GLOFs may coincide with monsoonal flash floods (Allen et al., 2016), and it remains difficult to distinguish the contribution of either type of flooding to observed damage. Landslides from undercut hillslopes may occur with a time lag to the outburst flood (Cook et al., 2018), so these damages may not be included in the initial estimate of damages. Many references therefore resorted to reporting only, if at all, the overall presence or absence of losses and damages."*

We will also add more information on regional differences in GLOF impacts to the manuscript as follows: *(L241-249): "Flood damages are mentioned for 404 GLOFs. Almost half of the GLOFs with reported damages were associated with ice-dammed lakes (49%), followed by moraine-dammed lakes (20%), and water pockets (17%) (**Fig. 8**). (…) The majority of GLOFs with reported damages occurred in HMA (34%), the European Alps (27%), and NW North America (22%). Hardly any societal impacts from GLOFs were reported in Greenland according to our database. (…) The most commonly reported impacts were destroyed bridges (n = 248), economic losses (n = 127), and damaged or debris covered roads (n = 104). (…) High Mountain Asia had at least 122 destroyed bridges, about half of the bridges that were globally reported to be destroyed by GLOFs. (…) Most GLOFs that caused economic losses or damage to bridges, buildings and roads, originated from ice-dammed lakes (**Fig. 8**). (…) In HMA, Scandinavia, Iceland, the Andes, and the European Alps, economic losses most commonly include agricultural damage, for instance the loss of crops, farmland, and cattle. In contrast, in the Pacific NW, economic losses mainly affect the touristic sector, for instance flooding or destruction of campgrounds. (…) Our data contain 44 deadly GLOFs, 29 of them with a reported number of victims, six known to have killed more than 100 people each. Many sources remained vague or offered estimates about the number of fatalities (e.g. Fushimi, 1985; Fort, 2015), mostly due to missing information. At least 33 GLOFs caused damage to utilities, for example by cutting off or shortening the local water supply, destroying pipes, or causing damage to hydropower plants. Most of the GLOFs that caused damage to utilities originated from moraine-dammed lakes (**Fig. 8**)"*

**References:**

Allen, S. K., Rastner, P., Arora, M., Huggel, C., and Stoffel, M.: Lake outburst and debris flow disaster at Kedarnath, June 2013: hydrometeorological triggering and topographic predisposition, Landslides, 13, 1479–1491, https://doi.org/10.1007/s10346-015-0584-3, 2016.

Cook, K. L., Andermann, C., Gimbert, F., Adhikari, B. R., and Hovius, N.: Glacial lake outburst floods as drivers of fluvial erosion in the Himalaya, Science, 362, 53–57, https://doi.org/10.1126/science.aat4981, 2018.

Emmer, A.: GLOFs in the WOS: bibliometrics, geographies and global trends of research on glacial lake outburst floods (Web of Science, 1979–2016), Nat. Hazards Earth Syst. Sci., 18, 813–827, https://doi.org/10.5194/nhess-18-813-2018, 2018.

Emmer, A., Allen, S. K., Carey, M., Frey, H., Huggel, C., Korup, O., Mergili, M., Sattar, A., Veh, G., Chen, T. Y., Cook, S. J., Correas-Gonzalez, M., Das, S., Diaz Moreno, A., Drenkhan, F., Fischer, M., Immerzeel, W. W., Izagirre, E., Joshi, R. C., Kougkoulos, I., Kuyakanon Knapp, R., Li, D., Majeed, U., Matti, S.,

Moulton, H., Nick, F., Piroton, V., Rashid, I., Reza, M., Ribeiro de Figueiredo, A., Riveros, C., Shrestha, F., Shrestha, M., Steiner, J., Walker-Crawford, N., Wood, J. L., and Yde, J. C.: Progress and challenges in glacial lake outburst flood research (2017–2021): a research community perspective, Nat. Hazards Earth Syst. Sci., 22, 3041–3061, https://doi.org/10.5194/nhess-22-3041-2022, 2022.

---

## Author Comment (AC3)

We thank the reviewer for their comprehensive and constructive comments on our work. Below, we respond to their comments in blue font and describe how we will address these comments in the revised manuscript in black font. References to specific lines refer to the initial manuscript.

**#Referee 3**

**R3C1:** The paper presents detailed database on the GLOF events, based on large number peer-review papers satellite and aerial image analyses and other sources. This is undoubtedly great contribution to the data collection on GLOF and should be published and be available to other researchers.

**R3A1:** We thank the reviewer for acknowledging the value of our database and the accompanying manuscript.

**General Comments:**

During reading the paper several questions arise:

**R3C2:** 1. When you write about standardized protocol for reporting characteristics of GLOF ("Our data collection process emphasizes the support of local experts in contributing previously undocumented cases, and we recommend applying systematic protocols when reporting new cases"), do you have some special questionnaire (protocol), that can be added to the paper? Or do you mean that it is recommended to other researchers just to collect the same 57 characteristics of GLOF as in your study? It is a little bit confusing.

**R3A2:** We regard our database as a suggestion for other researchers as to which information could be collected when reporting a GLOF. Our choice of our 57 parameters reflects a compromise between data availability, reporting trends, physical or risk-related relevance, and thus inadvertently contains subjective elements. No GLOF in our database has complete information on all these parameters, and it seems unlikely that any future appraisal will capture all this information for a given GLOF in the future, especially if they were detected retrospectively with remote sensing and lack ground truth.

Furthermore, we believe that the only way for implementation of a wide-ranging, ideally globally applied, standardized protocol is to go into discussion with other researchers in this specific field and make a recommendation based on a community perspective. We are going to attend the GSA Penrose conference (https://www.jsg.utexas.edu/penrose-2023/) soon, and hope to get feedback on essential variables that need to be gathered when observing new GLOFs or detecting previous GLOFs in hindsight (see **R1A3**).

**R3C3:** 2. I agree with referee 1 (Adam Emmer), that it is not clear, how can local researchers from different regions contribute data about new events to your database. Such additional option can do database more "active" and at the same time more sited.

**R3A3:** We would like to refer the reviewer to our reply **R1A3**, which we copy here for convenience:

We agree that active input from the research community is key to successfully maintaining our database and avoiding major data gaps. We followed the reviewer's idea and implemented a submission form on our website http://glofs.geoecology.uni-potsdam.de/ to ease the reporting of missing events for other researchers (see screenshot below). We will review each submitted case or reference individually and add them to the next version release of the database. To inform the reader about this option, we will add to the data availability section (L373): *"Our database is an ongoing project, and we offer a web-based, interactive map that grants access to the most recent state of the database (http://glofs.geoecology.uni-potsdam.de). This website includes a submission form that*

*enables the user to report missing or recently occurred GLOFs (**Fig. 12**).”*. Considering future updates, Zenodo will remain the version-controlled access to our database. Zenodo allows us to update our database under a new version using the same DOI, and we will release a version 3.1 as soon as a sufficient amount of new data has gathered. Furthermore, we plan to discuss schemes for standardized reporting and joint contribution within the GLOF community at an upcoming Penrose conference on outburst floods (https://www.jsg.utexas.edu/penrose-2023/).

[Figure]

**Figure 12: Submission form on** http://glofs.geoecology.uni-potsdam.de **that users can use to report missing or recently occurred GLOFs.**

**R3C4:** 3. Can you provide some additional explanations and examples about future prospections in the use of the database on the GLOF events? Now it is written in the conclusion only as "Following this approach, our collated database allows for objective comparisons on different spatial or temporal scales. Potential analysis based on the data might concern trends in GLOF occurrence, magnitude and impact, providing a valuable base for future hazard, risk assessment, and early warning". The rest is up to the reader's and user's imagination. But you are certainly have valuable experience with the previous version of your database and there are some examples in the literature, which can be added either to the introduction or to the conclusions.

**R3A4:** We thank the reviewer for pointing this out. We will extend the following paragraph in the conclusion as follows (L382): "*Following this approach, our collated database allows for objective comparisons on different spatial or temporal scales. Potential analysis based on the data might concern trends in GLOF occurrence, magnitude and impact, providing a valuable base for future hazard, risk assessment, and early warning. (…) Local analyses could address GLOF recurrence intervals to better reconcile them with design floods in river hydrology. On a global and regional scale, our database could*

*help quantify the impact of global warming on the frequency, timing and magnitude of GLOFs, and investigate links between population growth and reported GLOF impacts."*